# Enhancing the Quality of Life of Patients with Multiple Sclerosis: Promising Results on the Role of Cognitive Tele-Rehabilitation Plus Virtual Reality

**DOI:** 10.3390/brainsci13121636

**Published:** 2023-11-25

**Authors:** Maria Grazia Maggio, Antonino Cannavò, Angelo Quartarone, Alfredo Manuli, Paolo Tonin, Rocco Salvatore Calabrò

**Affiliations:** 1IRCCS Centro Neurolesi “Bonino Pulejo”, 98123 Messina, Italy; mariagrazia.maggio@irccsme.it (M.G.M.); angelo.quartarone@irccsme.it (A.Q.); 2A.O.U. Policlinico “G. Martino”, Via Consolare Valeria, 98124 Messina, Italy; antonino.cannavo.85@gmail.com (A.C.); manulialfredo@gmail.com (A.M.); 3S’Anna Institute, 88900 Crotone, Italy; patonin18@gmail.com

**Keywords:** cognitive rehabilitation, multiple sclerosis, quality of life, remote cognitive training

## Abstract

(1) Background: Patients with multiple sclerosis often face obstacles accessing traditional rehabilitation programs, primarily due to mobility limitations. Tele-rehabilitation (TR) is seen as a promising solution to overcome these barriers, though its precise influence on patients’ quality of life (QoL) has not been thoroughly investigated. Thus, the aim of the present study was to assess the feasibility of a cognitive TR in a sample of Italian patients with MS. (2) Methods: Thirty-six patients diagnosed with MS, attending the Robotic and Behavioral Neurorehabilitation Unit of the IRCCS “Bonino-Pulejo” Neurolesi Center in Messina, Italy, between October 2019 and March 2020 were enrolled in the study. All patients were randomly assigned, using block randomization with a block size of 2 × 2, to two groups: the control group (CG), composed of 16 patients who received traditional cognitive training, and the experimental group (EG), composed of 20 patients who underwent TR training with a VRRS (virtual reality rehabilitation system). Each patient underwent an assessment before (T0) and immediately after (T1) the rehabilitation treatment, using the Quality of Life-54 Multiple Sclerosis (MSQoL-54). (3) Results: Only in the EG, we observed a statistically significant improvement in the QoL related to mental well-being following the paired T-test (MSQoL *p*-value < 0.001). Notably, no significant differences were found in the CG (MSQoL *p*-value of 0.67). (4) Conclusions: Our data suggest that TR training combined with VR has the potential to improve the well-being of individuals with MS.

## 1. Introduction

Multiple sclerosis (MS) is an inflammatory disorder of the central nervous system, characterized by the loss of myelin in multiple areas [1]. The disease can onset at any age, but it most commonly affects individuals between 20 and 40 years of age, with women being twice as likely to be affected as men [2]. The most common form is marked by phases in which the disease presents symptoms, followed by periods of varying duration during which the symptoms are remitted (Relapsing–Remitting MS). In the early stages of the disease, the regression of symptoms is nearly complete, but over time, the symptoms persist for longer periods, leading to progressive disability and negatively affecting the quality of life (QoL) of the patients and their caregivers [3]. Cognitive impairments are evident in roughly half of individuals with MS. Neuropsychological studies demonstrate cognitive deficits in 40–65% of MS patients, particularly impacting memory, sustained attention, and information processing speed [4]. These cognitive challenges can hinder a patient’s ability to perform their work and engage in social activities, independently of their physical disabilities, thereby significantly affecting their overall QoL [5]. Engaging in rehabilitation training can serve as a mechanism to enhance stress resilience, support neural protection and regeneration, and mitigate long-term disability [6]. However, MS patients frequently encounter difficulties in accessing regular rehabilitation programs due to various barriers, including mobility limitations [7]. Tele-rehabilitation (TR) has emerged as a potential solution to address these challenges, even though its specific impact on the QoL has not been systematically examined [7].

Previous research has indicated the beneficial effects of rehabilitation in MS patients, ranging from traditional “paper and pencil” methods to more innovative approaches, such as virtual reality (VR) and tele-rehabilitation (TR) [6,7]. In this context, remote rehabilitation protocols offer a promising alternative to traditional in-person rehabilitation. TR addresses mobility-related issues and has the potential to reduce healthcare costs [7,8]. Furthermore, it is worth noting that prior studies have suggested that the clinical effectiveness of TR may be comparable to or even superior to traditional rehabilitation programs [7,8]. Various authors have also demonstrated that the benefits of TR are enhanced when used in conjunction with VR, which provides concrete, sustainable, and realistic experiences [6,7,8,9]. The simulation of real-life scenarios, through playful elements, fosters patient engagement and the potential for applying what has been learned in everyday life [6]. Thus, the aim of the present study was to evaluate the feasibility of a cognitive TR in a sample of Italian patients with MS.

## 2. Materials and Methods

### 2.1. Study Population

This exploratory study included thirty-six patients diagnosed with MS and attending the Robotic and Behavioral Neurorehabilitation Unit of the IRCCS Centro Neurolesi “Bonino-Pulejo” in Messina, Italy, between October 2019 and March 2020. All patients were randomly assigned, utilizing block randomization with a block size of 2 × 2, to two groups: the CG (control group), consisting of 16 patients who received traditional cognitive training, or the EG (experimental group), consisting of 20 patients who underwent TR training with VRRS (virtual reality rehabilitation system).

Inclusion criteria were (i) a diagnosis of MS following the most recent McDonald’s criteria [10]; (ii) patients who had been on stable therapy for a minimum of six months before entering the study; (iii) the presence of mild to moderate cognitive impairment, as determined by a Montreal Cognitive Assessment (MoCA) score between >18 and <27; and (iv) an EDSS (Expanded Disability Status Scale) score below 5.

Exclusion criteria were (i) individuals aged above 77 or below 18 years; (ii) the presence of severe medical or psychiatric conditions that might potentially interfere with the assessment or training; and (iii) a clinical and/or neuroradiological relapse of MS within the six months preceding enrollment.

### 2.2. Ethics

The study was conducted in accordance with the 1964 Helsinki Declaration. Each study participant provided informed consent.

### 2.3. Procedures

Each patient underwent an assessment before (T0) and immediately after (T1) the rehabilitation treatment, using the Quality of Life-54 Multiple Sclerosis (MSQoL-54). The MSQoL-54 is a questionnaire designed to evaluate the quality of life in individuals suffering from MS. This tool assesses various aspects of physical, emotional, and social well-being, offering insights into the impact of MS on a patient’s overall quality of life. With its 54 items, the MSQoL-54 covers several areas, including physical function, cognitive function, emotional well-being, energy, and general health. Patients rate their well-being in each area, providing information on the effects of the disease on their daily life and overall well-being.

All study participants received cognitive rehabilitation (CR) three times a week for eight weeks, i.e., 24 sessions, with each session lasting approximately 45 min. Each session aimed to stimulate specific cognitive domains. The complexity of the exercises increased according to individual progress, which was determined by successful responses (i.e., when 9 out of 10 answers were correct) and a low number of errors (less than one error). In summary, both groups underwent the same number of neurorehabilitation sessions, but only the experimental group (EG) received cognitive training using VRRS.

Specifically, the control group (CG) received traditional CR, which involved paper worksheets containing a variety of cognitive exercises and games (such as puzzles and memory tasks) designed to stimulate different cognitive abilities. The CG had to implement the provided exercises on their own. The type and difficulty level of exercises were systematically and randomly adjusted to prevent any learning bias. For the experimental group (EG), CR was carried out using the VRRS (Khymeia, Padua, Italy), an internationally patented Class I-certified medical device [11,12]. This system emphasizes user-friendliness, high customization capabilities, automatic reporting, and TR functions (Figure 1).

The VRRS offers a comprehensive suite of cognitive exercises, exceeding 50 in number, meticulously organized according to distinct cognitive functions. These functions include memory, attention, language, spatiotemporal orientation, planning, reasoning, and various executive functions, including calculation.

These cognitive exercises can be broadly classified into two primary categories:-2-D Exercises: Patients engage with objects and scenarios through either the touchscreen interface or a specialized magnetic tracking sensor paired with a compressible object. This setup effectively emulates mouse-like interaction skills, enhancing the overall user experience.-3-D Exercises: Within this category, patients immerse themselves in three-dimensional scenarios, interacting with virtual objects. This interaction is facilitated by a magnetic tracking sensor positioned above the hand, allowing for precise tracking of the 3-D position of the final effector.

The VRRS, with its emphasis on accessibility, adaptability, and automated progress tracking, emerges as a cutting-edge tool in the realm of cognitive rehabilitation.

In this research, the home-based TR system was used to facilitate a cognitive exercise program. The VRRS was designed to accommodate a wide range of patient-specific exercises, which were conducted with the remote guidance of an expert psychological therapist. The patient and the therapist met at the same time but in different places (patient at home and therapist in hospital), connected via the device. The neuropsychologists presented the patients with exercises, using which the patient performed rehabilitation on the device screen.

### 2.4. Statistical Analysis

The data were analyzed using SPSS version 16.0 (SPSS Inc., Chicago, IL,USA), with statistical significance set at *p* < 0.05. Descriptive statistics were presented as either mean ± standard deviation or median ± first–third quartile, depending on the nature of the data, while categorical variables were expressed as frequencies and percentages.

To assess the normality of the variables, the Kolmogorov–Smirnov test was employed. Given that the data subjected to the Kolmogorov–Smirnov test presented a normal distribution, a post hoc *t*-test for group differences in time and performance was conducted using Student’s *t*-tests, with the Bonferroni correction (0.01) applied. A paired t-test was utilized in the post hoc analysis to compare cognitive performance at T0 and T1.

## 3. Results

All patients completed the treatments without experiencing adverse effects. No drop-out was recorded. In total, thirty-six patients with MS were analyzed. For more details of the sample, see Table 1.

However, no significant statistical differences were detected between the two groups at baseline with regard to age (p = 0.88), gender (p = 0.35), and the MSQoL-54 in the physical (p = 0.36) and mental (p = 0.89) aspects.

Only in the EG, we observed a statistically significant improvement in the QoL related to mental well-being following the paired T-test. At time T0, the average MSQoL mental score was 56.4 ± 16.5, while at time T1, an average score of 73.0 ± 11.8 was recorded, with a *p*-value < 0.001. Notably, no significant differences were found in the CG (at time T0, the average MSQoL mental score was 57.1 ± 16.5, while at time T1, the average score was 55.5 ± 17.5, with a *p*-value of 0.67).

## 4. Discussion

In this study, we examined the influence of a home-based personalized exercise program (using the VRRS) on scores related to disease-specific QoL items. Our findings indicate that the intervention group, which received TR plus VR, experienced more substantial enhancements in QOL compared with the control group receiving conventional care. This study provides valuable insights into the benefits of TR training for MS patients, suggesting that the tool may lead to better mental well-being, a important aspect of the life of patients.

We selected TR combined with VR, using VRRS, as this innovative approach could have a greater impact on the cognitive improvement of these patients, favoring multidomain stimulation at the patient’s home and avoiding costs related to hospitalization and/or the effort of the journey to reach our center as outpatients [7]. It also encourages longer sessions using VR, resulting in more engaging training for the patient. VR has the potential to improve patients’ motivation and adherence to rehabilitation, leading to more extensive and intensive training [13]. It has been highlighted that a lack of motivation in rehabilitation settings can lead to poor adherence to prescribed exercise regimes and reduced training outcomes [14]. VR offers a solution by providing a stimulating and varied environment for the development of cognitive and motor skills during rehabilitation, potentially improving treatment plan adherence and the well-being of patients, as observed in our sample. Patients motivated by VR experiences may benefit from increased attention, possibly leading to better functional outcomes and improving neurotransmission systems. Indeed, VR training, with its multisensory feedback and patient engagement, can help mitigate the impact of brain injury in individuals with MS over time [6,9]. Moreover, exercises carried out in a virtual environment offer “enhanced feedback”, enabling patients to develop a better “awareness of the results” of their movements and the “quality of the movements” themselves, with evident benefits at the behavioral, cognitive, and motor levels [6]. In fact, the use of multisensory feedback and repeated task execution with sensory stimulation contribute to promoting brain plasticity processes. VR activates mirror neurons and integrates perception, cognition, and action, enhancing the effects of training and the patient’s sense of self-efficacy [15].

To sum up, this greater improvement in the EG is probably due to the role of VR in improving cognitive abilities, thus improving mental states and aspects related to QoL.

In recent years, there has been a growing emphasis on the concept of QoL in MS patients, as it is critical to recognize the psychosocial impact on individuals [16]. Indeed, patients’ perception of their physical and mental well-being significantly influences their autonomy, well-being, and potential functional recovery after rehabilitation [14]. Therefore, improving the perceived QoL is essential in performing and planning effective interventions for the patient.

However, this study has limitations due to the small sample size and lack of a follow-up period. The small sample size may not be sufficient to reliably support our findings. The sample size is limited due to various factors, such as participant availability and eligible resource constraints; however, our sample may be adequate to acquire preliminary insights and pilot data to carry out more extensive future studies.

Furthermore, the study included patients with MS without accounting for some factors, such as the age of the onset of the disorder. Lastly, there is an absence of follow-up data, as well as an examination of other indicators that could assess the impact of the training, as these factors are beyond the aim of this study. While the results are promising, further research with larger and more diverse samples is needed to validate the potential benefits of cognitive tele-rehabilitation combined with virtual reality for enhancing the well-being of individuals with multiple sclerosis.

## 5. Conclusions

In conclusion, our data suggest that TR training combined with VR has the potential to improve the well-being of individuals with MS, as confirmed by the higher scores in the MSQoL obtained by the EG. Further and larger multicenter studies are needed to confirm these promising findings and pave the way for the improvement of care of patients with MS.

## Figures and Tables

**Figure 1 brainsci-13-01636-f001:**
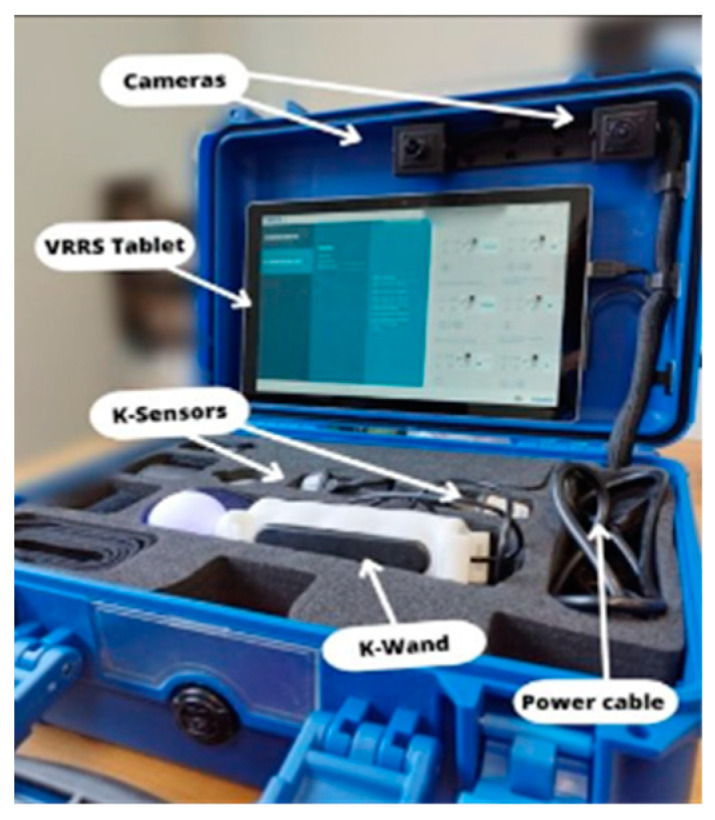
VRRS-HomeKit.

**Table 1 brainsci-13-01636-t001:** Demographic and clinical characteristics of the patients.

	Experimental	Control	All	*p*-Value
Patients	20	16	36	
Age	47.4 ± 10.2	50.5 ± 8.7	48.8 ± 9.6	0.88
Education	11.3 ± 3.3	11.6 ± 0.8	11.9 ± 4.9	0.11
Gender				0.35
Male	8 (30.0%)	6 (37.5%)	14 (38.9%)	
Female	12 (70.0%)	10 (62.5%)	22 (61.1%)	
Disease duration, (years)	10.7 ± 5.3	10.1 ± 6.1	10.5 ± 5.8	0.71
Median EDSS	4.7 ± 1.4	4.9 ± 0.4	4.8 ± 1.0	0.82

Mean ± standard deviation was used to describe continuous variables; proportions (numbers and percents) were used to describe categorical variables.

## Data Availability

Data are contained within the article.

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
