# Peer review of "Enhancing the Quality of Life of Patients with Multiple Sclerosis: Promising Results on the Role of Cognitive Tele-Rehabilitation Plus Virtual Reality"

_brainsci, 2023, doi:10.3390/brainsci13121636_

Round 1
Reviewer 1 Report
Comments and Suggestions for Authors
Thank you for submitting this interesting manuscript.
Having carefully reviewed your work, I would like to provide you with some feedback and suggestions for improvement.
1) Title: Add “cognitive” tele-rehabilitation to the title
2) Abstract line 16 + Introduction line 61: replace “evaluate” with “assess”.
3) Abstract: It is not clear how patients were assigned in line 20. Clarify please.
4) Abstract: You haven’t mentioned the outcome measures for QoL!
5) Introduction line 71: write the abbreviation between parentheses not the opposite.
6) Introduction line 76: write the abbreviation between parentheses not the opposite.
7) Introduction line 77: justify the age range you selected.
8) Results: You haven’t reported the patients’ characteristics especially the EDSS! I suggest creating a table with information about the EG, CG, and the whole sample statistics.
While the results are promising, further research with larger and more diverse samples is needed to validate the potential benefits of cognitive Tele-Rehabilitation combined with Virtual Reality for enhancing the well-being of individuals with multiple sclerosis.
Author Response
Dear Editor,
We would like to thank you and the reviewers for your interest in our study. We reviewed the manuscript as suggested as per his/her comments, using bold to highlight the changes.
To Reviewer 1: Having carefully reviewed your work, I would like to provide you with some feedback and suggestions for improvement.
Thank you for your feedback. We have implemented the suggested changes in the text to make it better.
1) Title: Add “cognitive” tele-rehabilitation to the title.
We thank the reviewer for the comment, we have changed the title as suggested.
2) Abstract line 16 + Introduction line 61: replace “evaluate” with “assess”.
We agree with the comment. We modified the text accordingly.
3) Abstract: It is not clear how patients were assigned in line 20. Clarify please.
We thank the reviewer for the request for clarification. We modified it accordingly.
4) Abstract: You haven’t mentioned the outcome measures for QoL!
Thanks for the suggestion, we have specified the QoL measures, i.e. MSQoL
5) Introduction line 71: write the abbreviation between parentheses not the opposite.
We have modified the abbreviation, as suggested.
6) Introduction line 76: write the abbreviation between parentheses not the opposite.
We have modified the abbreviation, as suggested.
7) Introduction line 77: justify the age range you selected.
The aim of rehabilitation is to observe the impact of TeleMS on an adult population, in our laboratory we do not deal with developmental age, furthermore excessively elderly people (over 77 years) are unable to use the system adequately.
8) Results: You haven’t reported the patients’ characteristics especially the EDSS! I suggest creating a table with information about the EG, CG, and the whole sample statistics.
We thank you for raising the issue. We added the Tables with the missing information
While the results are promising, further research with larger and more diverse samples is needed to validate the potential benefits of cognitive Tele-Rehabilitation combined with Virtual Reality for enhancing the well-being of individuals with multiple sclerosis.
We agree with your comments, and added this future prospective in the limitation of the study.
We agree with your comments and have added this future perspective to the limitation of the study.
The authors.
Reviewer 2 Report
Comments and Suggestions for Authors This an elegant exploratory study aiming to evaluate the efficacy (well-being) of web based cognitive rehabilitation in MS patients. The scientific interest is unquestionable and the manuscript is easy to follow. I do have the following points to be addressed: 1. In the methods: It is essential to provide more detailed information about the main tool of the research - the VRRS. 2. In the methods: It is important to provide more details about how the psychological therapist interacted with the patients 3. I would like to know the average duration of each virtual cognitive rehab section 1. In the methods: It is essential to provide more detailed information about the main tool of the research - the VRRS. 2. In the methods: It is important to provide more details about how the psychological therapist interacted with the patients 3. I would like to know the average duration of each virtual cognitive rehab section
Author Response
Dear Editor,
We would like to thank you and the reviewers for your interest in our study. We reviewed the manuscript as suggested as per his/her comments, using bold to highlight the changes.
To Reviewer :
This an elegant exploratory study aiming to evaluate the efficacy (well-being) of web-based cognitive rehabilitation in MS patients. The scientific interest is unquestionable and the manuscript is easy to follow.
Thank you for your feedback. We have implemented the suggested changes in the text to make it better.
I do have the following points to be addressed:
- In the methods: It is essential to provide more detailed information about the main tool of the research - the VRRS.
We thank the reviewer for focusing on this issue. We have made the description of the device clearer and more detailed (lines 117-131).
- In the methods: It is important to provide more details about how the psychological therapist interacted with the patients
We thank the reviewer for the request for clarification. We modified it accordingly (lines 137-140)
- I would like to know the average duration of each virtual cognitive rehab section.
Each group had the same number of sessions (as specified in lines 99-100), 24 sessions implemented three times a week, and each session (for both CG and EG) lasting about 45 minutes.
The authors.
Reviewer 3 Report
Comments and Suggestions for Authors
The main question addressed is the effectiveness of Tele-Rehabilitation (TR) combined with Virtual Reality (VR) in improving the quality of life (QoL) of patients with Multiple Sclerosis (MS), particularly focusing on mental well-being. The topic appears original and highly relevant. It addresses a specific gap in the field by exploring the combination of TR and VR in the rehabilitation of MS patients, an area not thoroughly investigated in previous research. This study adds to the subject area by demonstrating that TR combined with VR can lead to significant improvements in mental well-being of MS patients. This finding is significant as it suggests a new, effective approach to cognitive rehabilitation in MS.
Here is the list of suggested improvements:
- The authors should clarify the rationale behind their chosen methodology. This involves explaining why Tele-Rehabilitation (TR) combined with Virtual Reality (VR) was selected for this study, emphasizing the novelty and potential impact of this approach in cognitive rehabilitation for Multiple Sclerosis (MS) patients.
- Sample Size Justification:
- While the sample size may be limited, the authors can justify it by discussing the constraints they faced, such as availability of eligible participants or resources. They can also mention how this sample size is sufficient to achieve initial insights and pilot data for future, more extensive studies.
- The conclusions are consistent with the evidence presented. The study found a significant improvement in the QoL related to mental well-being in the experimental group, which aligns with their hypothesis about the benefits of TR combined with VR.
- The references seem appropriate and relevant to the study, covering aspects of MS, cognitive rehabilitation, and the use of VR in therapeutic contexts.
- The tables and figures adequately support the study's findings. They are well-integrated into the manuscript, providing clear and relevant information to complement the text.
Author Response
Dear Editor,
We would like to thank you and the reviewers for your interest in our study. We reviewed the manuscript as suggested as per his/her comments, using bold to highlight the changes.
To Reviewer:
The main question addressed is the effectiveness of Tele-Rehabilitation (TR) combined with Virtual Reality (VR) in improving the quality of life (QoL) of patients with Multiple Sclerosis (MS), particularly focusing on mental well-being. The topic appears original and highly relevant. It addresses a specific gap in the field by exploring the combination of TR and VR in the rehabilitation of MS patients, an area not thoroughly investigated in previous research. This study adds to the subject area by demonstrating that TR combined with VR can lead to significant improvements in mental well-being of MS patients. This finding is significant as it suggests a new, effective approach to cognitive rehabilitation in MS.
Thank you for your feedback. We have implemented the suggested changes in the text to make it better.
Here is the list of suggested improvements:
- The authors should clarify the rationale behind their chosen methodology. This involves explaining why Tele-Rehabilitation (TR) combined with Virtual Reality (VR) was selected for this study, emphasizing the novelty and potential impact of this approach in cognitive rehabilitation for Multiple Sclerosis (MS) patients.
Thank you for raising this issue. We have added this aspect in the discussion section (180-184).
- Sample Size Justification: While the sample size may be limited, the authors can justify it by discussing the constraints they faced, such as availability of eligible participants or resources. They can also mention how this sample size is sufficient to achieve initial insights and pilot data for future, more extensive studies.
Thank you for the review. We specified this issue in the limitations section (lines 211-216).
- The conclusions are consistent with the evidence presented. The study found a significant improvement in the QoL related to mental well-being in the experimental group, which aligns with their hypothesis about the benefits of TR combined with VR.
- The references seem appropriate and relevant to the study, covering aspects of MS, cognitive rehabilitation, and the use of VR in therapeutic contexts.
- The tables and figures adequately support the study's findings. They are well-integrated into the manuscript, providing clear and relevant information to complement the text.
We thank the reviewer for the comments.
The authors.
Round 2
Reviewer 3 Report
Comments and Suggestions for Authors
I would like to thank the Authors for all corrections they have done.
I have no further comments.